# Therapeutic Exercise Parameters, Considerations and Recommendations for the Treatment of Non-Specific Low Back Pain: International DELPHI Study

**DOI:** 10.3390/jpm13101510

**Published:** 2023-10-19

**Authors:** Zacarías Sánchez Milá, Teresa Villa Muñoz, María del Rosario Ferreira Sánchez, Raúl Frutos Llanes, José Manuel Barragán Casas, David Rodríguez Sanz, Jorge Velázquez Saornil

**Affiliations:** 1NEUMUSK Group, Facultad de Ciencias de la Salud, Universidad Católica de Ávila, 05005 Ávila, Spain; zacarias.sanchez@ucavila.es (Z.S.M.); mrosario.ferreira@ucavila.es (M.d.R.F.S.); jmanuel.barragan@ucavila.es (J.M.B.C.); 2FisioSalud Ávila, 05002 Ávila, Spain; 3Facultad de Enfermería, Fisioterapia y Podología, Universidad Complutense de Madrid, 28040 Madrid, Spain; davidrodriguezsanz@ucm.es

**Keywords:** Delphi study, low back pain, exercise, physiotherapy techniques, musculoskeletal disease

## Abstract

Background: Therapeutic exercise (TE) recommendations for non-specific low back pain (LBP) are meant to support therapy choices for people who suffer from this condition. The aim of this study was to reach an agreement on the definition and use of TE in the care of people with LBP. Methods: A Delphi study was carried out with a formal consensus procedure and sufficient scientific evidence, using an established methodology. Four rounds of anonymous questionnaires were administered to create useful suggestions and instructions in terms of the therapeutic activity for patients with LBP, and a group consensus conference. Results: A consensus was reached on most of the questions after 35 physiotherapists completed the questionnaires. Participants agreed that proper TE requires correct posture, body awareness, breathing, movement control, and instruction. Patients with LBP were advised to participate in supervised sessions twice a week for 30 to 60 min for a period of 3 to 6 months. Participants added that tailored evaluation and exercise prescription, monitoring, and functional integration of exercise, as well as using specific equipment, would benefit patients with LBP. Conclusions: TE recommendations for patients with LBP should be dosed and customized based on their personal psychological needs, level of fitness, and kinesiophobia.

## 1. Introduction

In order to be classified as low back pain (LBP), one must experience pain below the final set of ribs and above the buttock [1]. It is estimated that 80% of adults will experience LBP at some point in their lives [2,3,4]. LBP is a significant global public health issue. In recent years, it has been the leading cause of absence from work and medical rehabilitation needs. LBP is just one step behind mental health as a reason for early disability-based retirement [3,4,5,6]. In Germany, the disease management guidelines for non-specific LBP have been modified, stressing psychosocial workplace variables, early multidisciplinary therapy, and placing exercise ahead of bed rest [6]. Risk factors, such as certain regular postures that produce deviations, excess weight, and abdominal wall distension, may facilitate the development of non-specific LBP [3,7].

Only a small proportion of the population recognizes the pathological reason for their LBP, with 90% of cases being non-specific [2,3,4,5,6] and having an unknown medical cause [1].

In accordance with studies, the prevalence of non-specific LBP is 84% worldwide [1], increasing between the ages of 60 and 65, and decreasing steadily thereafter [6]. LBP is one of the main causes of disability worldwide, accounting for 54% of the increase in disability between 1990 and 2015 in low- and middle-income countries [8].

As some authors have stated, the higher prevalence of non-specific LBP in women than in men may be due to anatomical and functional differences. Women are smaller and have lower bone density, less muscular mass and weaker joints [9].

Patients frequently suffer from recurrent episodes of non-specific LBP. The annual incidence rate is higher in the third decade [2,3], ranging from 15% to 45%. Various medical and physiotherapeutic treatments are available, from the recommendation of analgesic measures to more conventional measures, such as passive therapies, including massage therapy and the application of electrotherapy for analgesic purposes. Nowadays, however, we are facing a paradigm change in which treatment is much more active on the patient’s side, and the term “hands off” has even been coined, referring to the previously explained concept of active therapy [7]. This new therapy is a therapeutic exercise (TE) tailored to the individual patient, in terms of intensity, the patient’s own pain and dosage. Physiotherapists are trained to treat various musculoskeletal complaints by means of TE and can prescribe such exercises [10].

It has been shown that patients with non-specific LBP experience less pain and disability when engaging in therapeutic activity [11,12,13]. According to recent searches, improvements are comparable regardless of the workout type [14,15,16]. When suggesting an exercise program for people with non-specific LBP, it is advisable to consider the logic behind TE approaches [14]. With this method, workout plans can be individually tailored for optimal effectiveness. Additionally, workouts that target postural control and trunk muscular stabilization may be advantageous for patients with non-specific LBP [12,13,14]. The evidence on the effectiveness of TE in patients with non-specific LBP, as claimed by some meta-analyses and systematic reviews, is, however, conflicting [14,15]. This result was caused by a lack of studies, varying methodological quality and small sample sizes [14]. As it would not be appropriate to combine the findings of these studies in a meta-analysis, the heterogeneity of the primary studies in terms of demographics, interventions, comparisons and outcome measures further restricts the robustness of the research findings.

To reach an agreement on the definition and use of TE to treat patients with LBP, a Delphi survey of a group of European physiotherapists was performed. The results of this study will help researchers to plan upcoming TE studies and to interpret previous research [16]. The Delphi survey’s study questions were:What are your qualifications in relation to people with non-specific LBP?What is the best TE design for patients with non-specific LBP in terms of guidelines, level of supervision and equipment?Which guidelines are used to ensure that TE is prescribed and progressed safely for patients with non-specific LBP?

## 2. Materials and Methods

An international group of traumatologists, orthopedic surgeons, fundamental scientists, physical activity and sports scientists, and surgeons with experience treating non-specific LBP attended a meeting. A formal consensus procedure was conducted with the use of a verified methodology (consisting of four rounds of questionnaires administered to a set of subject matter experts, conducted anonymously and not coinciding with each other) [17]. We analyzed the available research, convened a consensus group meeting to formulate recommendations and then organized a wider consultation meeting with an open invitation for final endorsement. With the help of local, national and international experts in non-specific LBP, we performed iterative consensus research (Delphi). The Delphi method is classified as one of the general foresight procedures aiming to obtain the consensus of a group of experts based on the analysis and reflection of a defined problem [17]. The members of this group were recruited using specific expressions of interest and invitations from experts and four rounds of anonymous questionnaires for the Delphi study. Rounds 1 and 2 consisted of the creation and ranking of a long list of potential traits, while in rounds 3 and 4 the participants were asked to agree on a set of preliminary criteria after being informed of the results of the previous rounds. Most of the participants (72%) were highly qualified and skillful European clinical volunteers in LBP management, reflecting different levels of clinical experience. Three levels of assurance were incorporated into the preliminary criteria from the earliest rounds: TE therapy is ideal, useless, or irrelevant. In the fourth round, consensus was reached with extremely high levels of agreement (>89%) amongst all levels of criteria and subcategories. Overall, 96% of the panelists agreed that the criteria should be adopted. The NEUMUSK research group of the Catholic University of Ávila, which designed the study, was in charge of supervising the correct methodological use of the Delphi Consensus at all times and was responsible for the storage and custody of the study results. In addition, it reviewed and approved the rules, followed by its experts in physiotherapy, along with other specialists in different fields such as physical activity and sports sciences, traumatology and orthopedics.

### Recruitment

Participants were selected by means of purposive sampling, whereby a panel of “experts” were selected on the basis of their knowledge and experience of the topic, their availability and interest and skills to communicate. This selection method ensures that the results of the Delphi survey are based on informed opinions and that maximum participation rates are achieved. Snowballing techniques were also used to identify potential panel members. Snowballing techniques involve participants nominating or recommending others to participate in the study based on knowledge of the study’s inclusion criteria. The use of snowball recruitment techniques can increase both the size and diversity of the sample population. The recruitment process began with the principal investigator sending an email invitation to physiotherapists, physicians and physical activity and sport science experts who were likely to meet the selection criteria. This email included information about the research project and informed consent and screening forms. Participants were invited to contact the principal investigator by email or telephone to discuss the project. Participants were also encouraged to forward the project information to other interested professionals they thought might meet the selection criteria. Interested participants then emailed their completed screening and consent forms to the principal investigator. Once the screening and consent forms were received and checked, participants were formally included in the study. In the end, 35 participants made up the group of experts, as shown in Figure 1. The Delphi survey involved electronic questionnaires provided over the course of 5 months (May–September 2023). Participants were emailed electronic links to each questionnaire and received individual login details to complete their answers. Individual login details ensured the security of information and prevented duplicated responses. Participants were requested to complete each questionnaire within 2 weeks.

Responses to open-ended questions in the first questionnaire were summarized qualitatively using thematic analysis. Several researchers were involved in this process to ensure the validity and consistency of the approach. Themes identified from participant responses then were translated into statements about TE and people with LBP. These statements were utilized in the development of the next questionnaires.

Participants were requested to rank their level of agreement with a number of statements regarding TE in people with LBP using a 6-point Likert response scale (“strongly agree”, “agree”, “somewhat agree”, “somewhat disagree”, “disagree”, and “strongly disagree”). A 6-point Likert scale was selected because it has been shown to be valid, reliable, and suitable for use with educated individuals.

The Likert scale of responses was used to identify areas of consensus or non-consensus among the expert panel members. Prior to the commencement of this study, consensus was defined as when 70% to 100% of the panel members strongly agreed, agreed, or somewhat agreed (or strongly disagreed, disagreed, or somewhat disagreed) with an item. If the percentage of agreement or disagreement was less than 60%, however, it was concluded that a consensus had not been reached. Open-ended questions were also provided to ensure participants were able to express any further thoughts or opinions.

Below are the questionnaires and various questions sent to professionals to carry out the study using the Delphi Consensus methodology on the management of non-specific LBP pain using TE (questions 1 to 60 and tables from Appendix A). 

## 3. Results

The 35 specialists who participated in this study came to an agreement in the fourth round. TE is recommended as a course of treatment following a literature review and several multidisciplinary group sessions. The four expert rounds can be seen in Figure 2.

After four questionnaires, 91.7% (176/192) of the questions had consensus levels of agreement. However, 8.3% (16/192) of the questions could not be agreed upon. The components of consensus and non-consensus related to this study’s research topics are listed below.

What does “therapeutic exercise” mean in terms of those suffering from non-specific LBP?

From the questions regarding the definition of TE qualities, it was agreed that body awareness, breathing, control, education, individually adapted exercises, movement control and posture were identified as particularly significant elements of TE, specifically by 97.1% (33/34) of participants. 

Overall, 78.9% (15/19) of the critical elements of the TE protocol for patients with non-specific LBP were planned. The use of encouragement and feedback from the therapist, the functional integration of TE principles, the incorporation of home exercises, patient self-consciousness, and therapist reassessment were essential elements. Regarding the prescription of a specific number of exercises and the integration of resting and cooling activities, no agreement was reached.

Concerning the suitable parameters of TE and supervision for patients with non-specific LBP, an agreement was achieved within a range of values on 100% of the questions. Participants overwhelmingly agreed that supervised exercise sessions for patients with non-specific LBP should last between 30 and 60 min (100% agreement), should be performed twice a week (73.3% agreement), and should be completed within a period of 3 to 6 months (83.4% agreement).

These criteria were established, according to participant feedback, to make sure that clients remembered their exercises, used proper form, successfully corrected their motor patterns, strengthened their weak muscles, and accomplished their functional objectives. These guidelines also aimed to increase client satisfaction, motivation, and adherence within the existing constraints of availability and budget (100% agreement), as well as to enable the reduction, prevention, and self-management of symptoms and avoid frightened behavior.

One client per therapist was the suggested level of supervision by participants at the beginning of the program (80% agreement), and two to four clients per therapist after two weeks (100% agreement). Overall, 100% of participants believed that these degrees of supervision allowed for individualized exercise prescription, technique progression and monitoring, and ensured client self-care, pain and injury prevention, and a gradual decline in therapist dependence.

Furthermore, TE can supplement home activities, provide opportunities for growth, and offer customizable resistance. All questions relating to the customization of programs for people with non-specific LBP were agreed upon by the participants. Client goals, functional requirements, irritation, specific movement or activity anxieties, and body awareness are factors that should be given particular attention. A unanimous decision was obtained by all participants on all issues referring to the progression of exercise for those with non-specific LBP. The consensus among the participants was that the evolution of TE should have three main components: an increase in exercise complexity, a recreation involving a functional sport and the integration of exercise concepts.

The concept of prescribing TE to people with non-specific LBP was the subject of consensus on 94.7% (18/19) of the questions. Conducting an initial assessment, educating patients about the benefits of TE and chronic pain mechanisms, prescribing functionally relevant exercises in accordance with the client’s needs, ability, irritability, and pathology, supervising sessions, checking the effectiveness of the technique, encouraging breathing with movement, questioning belief systems to avoid fear, and routine reassessment of symptoms and functional outcomes were among the principles of great importance. Concerning the physical condition of patients, psychological state, and status in relation to kinesiophobia, no agreement was reached.

A summary of the decisions agreed upon by the experts after the four rounds can be seen in Figure 3.

## 4. Discussion

In this Delphi study, 35 health professionals agreed on most of the practical and definitional aspects of TE for people with non-specific LBP (Tabs 1). For 91.7% (176/192) of the items, consensus levels of agreement were attained after three rounds of questionnaires. The identification of TE features (1/34), must-have TE elements (4/19), essential equipment types (9/28) and their rationale for usage (1/11), and the basis of exercise prescription (1/19) were some of the points with non-agreement.

Participants concurred that all seven TE components—breathing, posture, flexibility, movement control, strength, core stability, and a mind–body connection—were appropriate for patients with non-specific LBP. These components were found in a recent systematic review of the literature [16]. The high median agreement reflected the importance placed on breathing, movement control, and posture. However, further investigation is needed to determine the relative importance of other distinctive qualities and vital elements.

Specific recommendations for the use of TE in the treatment of patients with non-specific LBP are provided by the consensus conclusions. The duration and frequency of TE sessions have been appropriate in view of these characteristics, but the length of exercise programs (i.e., 6–8 weeks) has often been inadequate [16,18]. Exercise trials for patients with non-specific LBP may find that the total number of sessions and hours of exercise are related to the effect size, so it may be important for future studies to make sure that TE interventions last between three and six months in order to achieve the best results [19].

The consensus conclusions also offer recommendations for the necessary tools and levels of supervision for applying TE to treat patients with non-specific LBP. The majority of TE trials for people with non-specific LBP have not used outside materials in their programs [20,21,22,23]. However, future research should examine the advantages of programs with and without the use of outside resources (use of materials such as elastic bands, rollers, etc.) considering the survey results. In future studies, grades of supervision should also be carefully considered because they could affect how well exercise works for people with non-specific LBP.

The guidelines for prescribing therapeutic exercise, which are similar to other exercise regimens that are successful in treating patients with non-specific LBP, were agreed upon by the participants. Participants, for instance, agreed that exercises should include stretching and strengthening and be individually designed and monitored [24,25,26]. Moreover, therapeutic activities should emphasize the coordination, strength, and endurance of the trunk muscles, respect clients’ treatment preferences (in the case of kinesiophobia) and incorporate cognitive behavioral therapy [27]. Additional clinical research is required to confirm the significance of further parts of the consensus related to individualization, prescription, and progression of workouts.

In published studies of participants with non-specific LBP, principles of therapeutic exercise, such as pelvic scale, concentration, and precision, were not discussed, indicating that they may not be significant according to our systematic analysis of the literature [15,16]. However, the principles of attention, accuracy, flow, pelvic scale, control, and breathing were taken into consideration while looking at consensus conclusions regarding the identification of TE features [27,28,29,30]. Although the CORE activation via the pelvic scale was the premise most frequently mentioned, high-intensity intervallic exercise for chronic pain should also be considered [31]. 

There are intrinsic limits to the Delphi method itself. Even if participants do not immediately contact each other, the iterative and anonymous group feedback process may persuade participants to agree. Bias among participants and researchers may result from this procedure. The results from Delphi surveys are only admissible as professional opinions and are ranked lower than primary studies in the hierarchy of evidence. A consensus of results does not necessarily imply that the group’s assessment is accurate. Therefore, these results need to be verified and put to the test in other clinical studies. 

Only 35 experts participated in this Delphi survey, which means that findings may be skewed, as only a proportion of physical therapists, physicians and physical activity and sports professionals experienced in the use of TE in people with LBP gave their opinion. Selection and response biases are likely to be present where physical therapists, physicians and physical activity and sports professionals who met the selection criteria were not invited to participate, did not agree to participate, or did not follow through in completing questionnaires.

## 5. Conclusions

The experts in the study concluded that TE approaches for patients with non-specific low back pain should be dosed and personalized according to the patient’s psychological needs, fitness level and kinesiophobia. In addition, they should be adapted to the level of pain of the patients, should be professionally supervised exercises lasting 30 to 60 min, twice a week and for at least 3 to 6 months. These results help us to understand how health professionals treat people with non-specific low back pain through TE. Future research on TE will benefit from this information, although it is important to evaluate the results in light of the limitations of the study.

## Figures and Tables

**Figure 1 jpm-13-01510-f001:**
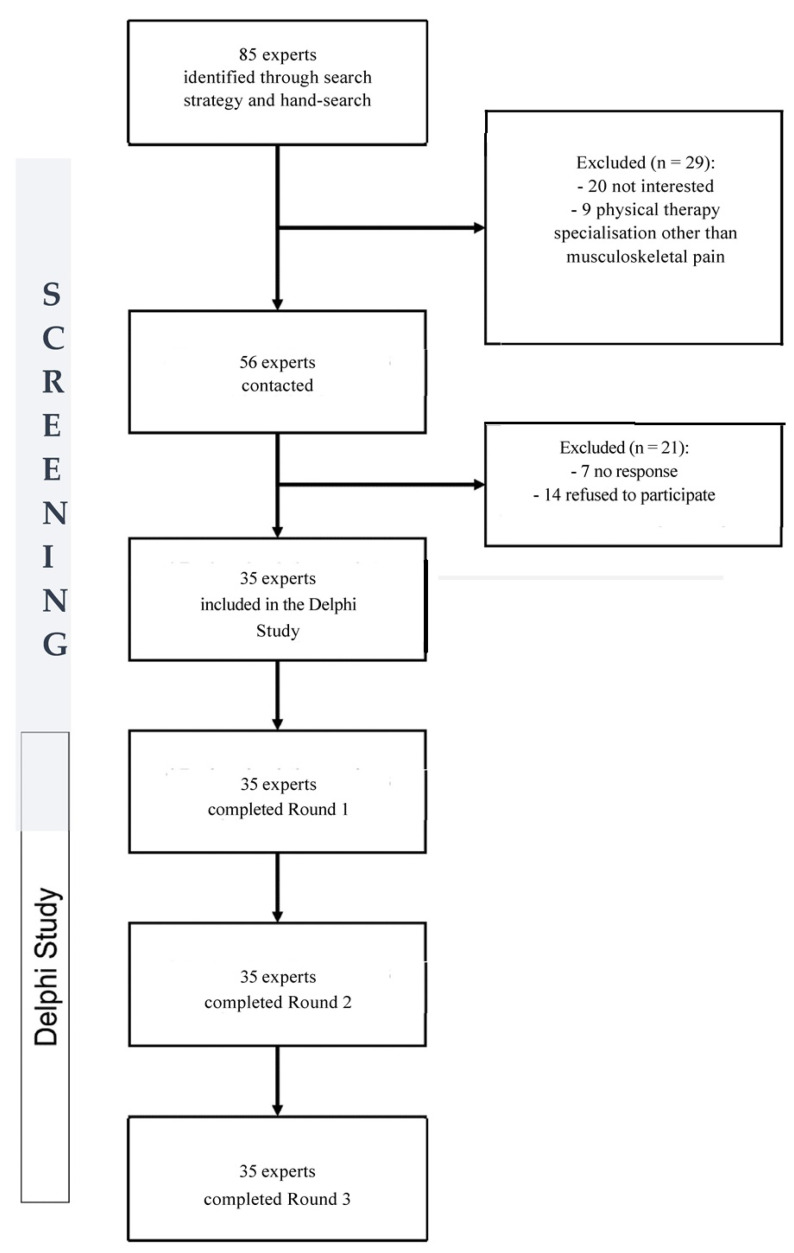
Expert recruitment diagram.

**Figure 2 jpm-13-01510-f002:**
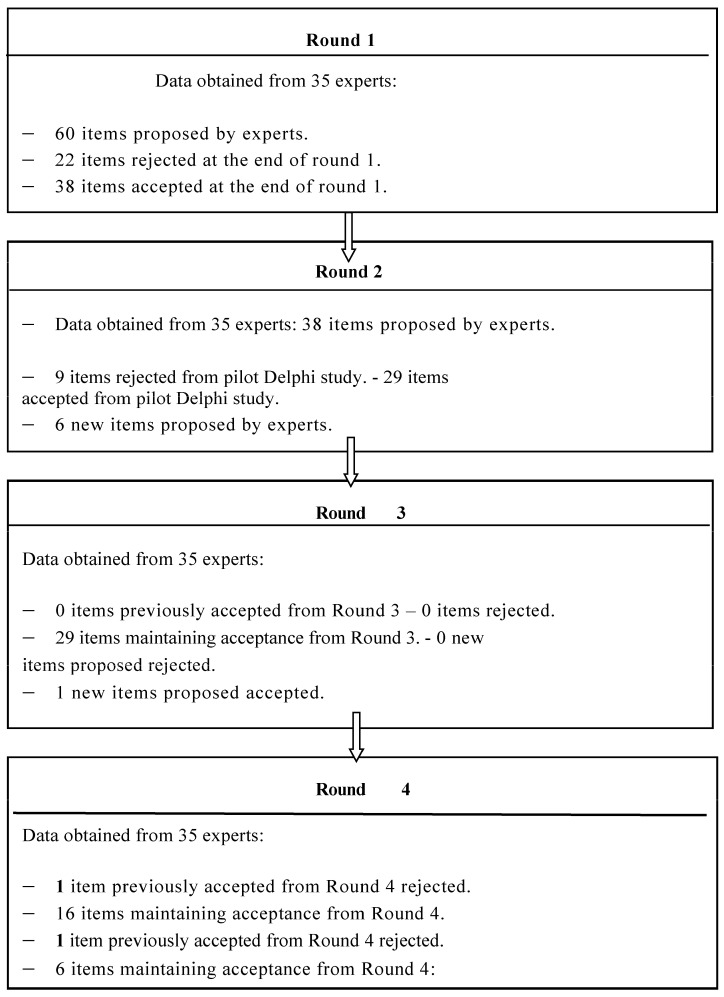
Consensus and four expert rounds.

**Figure 3 jpm-13-01510-f003:**
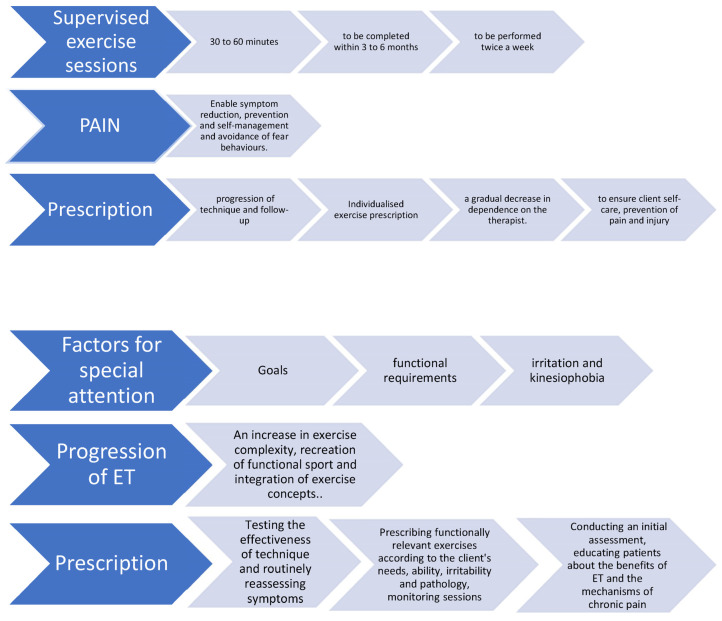
Agreements of the experts after the 4 rounds.

## Data Availability

Not applicable.

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
