# Peer review of "Therapeutic Exercise Parameters, Considerations and Recommendations for the Treatment of Non-Specific Low Back Pain: International DELPHI Study"

_jpm, 2023, doi:10.3390/jpm13101510_

Round 1

Reviewer 1 Report

Thank you for the authors for presenting this manuscript. At first I was keen at reviewing it, but upon review the manuscript was not polished enough nor presented in an easily read manner for me to fully review it. For example, it seemed some of the results were missing (eg a whole table of results). I have attached some comments of the sections I had read (part of the results and sections beforehand). I suspect the discussion would also need improvements before submission. 

Author Response

Thank you for the authors for presenting this manuscript. At first I was keen at reviewing it, but upon review the manuscript was not polished enough nor presented in an easily read manner for me to fully review it. For example, it seemed some of the results were missing (eg a whole table of results). I have attached some comments of the sections I had read (part of the results and sections beforehand). I suspect the discussion would also need improvements before submission. 

Reviewer 2 Report

Line 91 of Materials and Methods is not very clearly written.

Line 95 “a verified methodology” What is this methodology? Specify references.

Line 95  "We looked over the available research..." According to what criteria were the studies selected? What was the number of studies?

Line 100, describe the questions presented in the questionnaires.

Line 109, NEUMUSK is mentioned for the first time in the text, so you must write the full name.

Line 247, CORE activation via the pelvic scale.... Is this an abbreviation or did you mean "core activation"?

Author Response

Line 91 of Materials and Methods is not very clearly worded. The text has been modified in order to bring clarity to the manuscript, thank you for your comment.

Line 95 "a verified methodology" What is this methodology? Specify the references. Delphi consensus is an expert consensus method and a specific reference has been added, thank you.

Line 95 "We reviewed the available research..." According to what criteria were the studies selected? What was the number of studies? A literature search was carried out, but is omitted for reasons of length, thank you.

Line 100, describe the questions presented in the questionnaires. An appendix has been added to the manuscript, thank you.

Line 109, NEUMUSK is mentioned for the first time in the text, so write the full name. It is an acronym referring to the research group, it has been explained in the text, thank you.

Line 247, activation of the CORE via the pelvic scale.... Is this an abbreviation or did you mean "core activation"? Core Activation, thanks!

Reviewer 3 Report

Dear Authors,

The aim of your manuscript is quite interesting since a lot of effort is still needed to provide the best evidence-based care to patients suffering from low-back pain.

Anyway, I have several concerns about your work. Please follow my suggestion and try to improve the manuscript as best as you can.

1) Introduction

-Please explain acronyms at their first appearance in the text (es. line 31 "LBP")

- Line 35: put a reference at the end of the line

- Lines 43-45: redundant information

- Line 62: reference needed ("...previously explained concept of active therapy")

- Line 68: references should be put in square brackets

2) Materials and Methods

- Please explain what the Delphi methodology is and report an adequate number of references

- Lines 92-93: what do you refer to with "verified methodology"? Please explain it and put a reference

- Some lines should be totally rephrased (es. lines 99-100)

- "The creation and ranking of a lengthy list of potential traits made up Rounds 1 and 2." This list should be reported somewhere in the text or as an Appendix.

- "Participants were asked to agree on a set of preliminary criteria in rounds 3 and 4 after being informed about the outcomes of the previous rounds". Which are these "preliminary criteria"?

- The majority of the participants were highly skilled European clinical volunteers, reflecting various levels of clinical expertise. Please report the exact percentage when referring to "majority". Furthermore, "highly skilled" should be clearly explained (es. skilled in what?)

- Three levels of assurance were incorporated into the preliminary criteria from the earliest rounds: TE therapy is ideal, not useful, or indifferent. Please rephrase. 

- Please explain what the NEUMUSK is

3) Results

- The "components" should be highlighted. I suggest to divide the section into sub-paragraphs, one for each research question

- Line 154: Table 1 regards the experts' recruitment. To which table were you referring to? 

- Line 156: plese explain that "particularly crucial" giving an exact numeric definition

- Lines 158-160: exact numeric values should be reported for each single "element"

- Lines 225 and 226: what did you mean with "outside resources"?

- Lines 234: what did you mean with "the direction preferences of the clients"?

The same thing should be done for each further research question

4) Discussion

- Lines 203-206: all these "crucial" elements should be reported in the Results section, as already reported above

- Lines 206: the topics on which there eas or there was not a reached agreeement should be reported in Table 2

N. B. You should cite this article: Botta RM, Palermi S, Tarantino D. High-intensity interval training for chronic pain conditions: a narrative review. J Exerc Rehabil. 2022 Feb 24;18(1):10-19. doi: 10.12965/jer.2142718.359. PMID: 35356137; PMCID: PMC8934613. 

An extensive English language revision is needed for the entire manuscript. I strongly suggest to let an English native speaker revise your manuscript. Some lines are really badly written (es. lines 43-44 "..characterised by this [1]" or line 55 "Recurrent episodes of LBP are common in sufferers", lines 69-70, etc). 

Author Response

Dear Reviewer:

The following is how to proceed with your recommendations, thank you very much for your comments and constructive criticism.

Reviewer#1:

The aim of your manuscript is quite interesting since a lot of effort is still needed to provide the best evidence-based care to patients suffering from low-back pain.

Anyway, I have several concerns about your work. Please follow my suggestion and try to improve the manuscript as best as you can.

1) Introduction

-Please explain acronyms at their first appearance in the text (es. line 31 "LBP") The acronyms were previously inserted in the abstract, but we have decided to add it also in the first term of the introduction, as you suggested, thank you.

- Line 35: put a reference at the end of the line. Your comment has been taken into account and a quote has been added at the end of the line.

- Lines 43-45: redundant information. Your comment has been taken into account, thank you.

- Line 62: reference needed ("...previously explained concept of active therapy"). Your comment has been taken into account and the reference [7] on active exercise in low back pain has been added, thank you.

- Line 68: references should be put in square brackets. Your comment has been taken into account and the reference has been put in square brackets, thank you.

2) Materials and Methods

- Please explain what the Delphi methodology is and report an adequate number of references. The Dephi consensus definition and a recent quote on this concept have been added, thank you.

- Lines 92-93: what do you refer to with "verified methodology"? Please explain it and put a reference. This term refers to a way of proceeding with four rounds and a set of subject matter experts that is conducted anonymously and do not coincide with each other. It is a methodology already verified and carried out on numerous occasions by other scientists. Your comment has been taken into account and information has been added, thank you.

- Some lines should be totally rephrased (es. lines 99-100). Your recommendation has been taken into account and the commented lines have been modified, thank you.

- "The creation and ranking of a lengthy list of potential traits made up Rounds 1 and 2." This list should be reported somewhere in the text or as an Appendix. Appendix 1 has been added so that all information from the questionnaires can be verified.

- "Participants were asked to agree on a set of preliminary criteria in rounds 3 and 4 after being informed about the outcomes of the previous rounds". Which are these "preliminary criteria"? Appendix 1 has been added so that all information from the questionnaires can be verified.

- The majority of the participants were highly skilled European clinical volunteers, reflecting various levels of clinical expertise. Please report the exact percentage when referring to "majority". Furthermore, "highly skilled" should be clearly explained (es. skilled in what?). Your comment has been taken into account and information has been added (lines 106-108), thank you.

- Three levels of assurance were incorporated into the preliminary criteria from the earliest rounds: TE therapy is ideal, not useful, or indifferent. Please rephrase.

- Please explain what the NEUMUSK is. NEUMUSK is a research group of the Catholic University of Avila that has designed the study, and has been in charge of supervising at all times the correct methodological use of the Delphi Consensus and has been responsible for storing and safeguarding the results of the study. This information has been added in the manuscript, in a light-hearted manner, for the reader's information (lines 112-117). Thank you.

3) Results

- The "components" should be highlighted. I suggest to divide the section into sub-paragraphs, one for each research question

- Line 154: Table 1 regards the experts' recruitment. To which table were you referring to? It is incorrect, it has been modified, thank you.

- Line 156: plese explain that "particularly crucial" giving an exact numeric definition. Your comment has been taken into account and the entire paragraph has been modified (lines 161-164). Thank you.

- Lines 158-160: exact numeric values should be reported for each single "element"

- Lines 225 and 226: what did you mean with "outside resources"? It has been explained in parentheses in lines 234-235 in case it was not entirely clear, thank you.

- Lines 234: what did you mean with "the direction preferences of the clients"? Your comments have been taken into account and modified in lines 243 and 244, thank you.

The same thing should be done for each further research question

4) Discussion

- Lines 203-206: all these "crucial" elements should be reported in the Results section, as already reported above. Appendix 1 has been added so that all information from the questionnaires can be verified.

- Lines 206: the topics on which there eas or there was not a reached agreeement should be reported in Table 2. A new table (Table3) has been added so that criteria with a high percentage of agreement among experts can be checked. Thank you.

  1. B. You should cite this article: Botta RM, Palermi S, Tarantino D. High-intensity interval training for chronic pain conditions: a narrative review. J Exerc Rehabil. 2022 Feb 24;18(1):10-19. doi: 10.12965/jer.2142718.359. PMID: 35356137; PMCID: PMC8934613. Your comments have been taken into account and the reference you provided has been added, thank you.

Comments on the Quality of English Language:

An extensive English language revision is needed for the entire manuscript. I strongly suggest to let an English native speaker revise your manuscript. Some lines are really badly written (es. lines 43-44 "..characterised by this [1]" or line 55 "Recurrent episodes of LBP are common in sufferers", lines 69-70, etc). Your comment has been taken into account and the manuscript has been reviewed by an expert, thank you very much.

Round 2

Reviewer 1 Report

Thank you for the authors for the quick turnaround and responding to the specific items that were lists. On brief review, it seems some areas have improved (eg intro, methods), but firstly there are still major questions about how the authors have conducted the research/methods. Unfortunately, I do not have the time to review and highlight every single process/step missing, nor should require to edit the manuscript to a polished state for the authors, especially in a voluntary capacity. 

Another area that has deteriorated is the results - why dump/insert the whole Delphi questionnaire (approx 13-14 pages)without description. It doesn't necessarily add to the paper. The addition of table 24 is not discuss adequately nor makes sense. Why is there arrows? What are they for? Aren't they separate ideas so how do the arrows link these ideas? 

I feel that major rewrites will be necessary before it is up to an acceptable level for publication. 

Nil major issues on brief read. 

Author Response

Dear Reviewer 1:

Thank you very much for your review and your time. We have carried out a thorough restructuring on the manuscript and your comments have been taken into account, you can check it in the attached document. 

Best regards and thank you.

Below, we explain the changes that have been made based on your feedback:

Thank you for the authors for the quick turnaround and responding to the specific items that were lists. On brief review, it seems some areas have improved (eg intro, methods), but firstly there are still major questions about how the authors have conducted the research/methods. Unfortunately, I do not have the time to review and highlight every single process/step missing, nor should require to edit the manuscript to a polished state for the authors, especially in a voluntary capacity. Your comment has been taken into account and this section has been significantly restructured and expanded. Let us hope that this time we will meet your criteria. Best regards.

Another area that has deteriorated is the results - why dump/insert the whole Delphi questionnaire (approx 13-14 pages)without description. It doesn't necessarily add to the paper. The addition of table 24 is not discuss adequately nor makes sense. Why is there arrows? What are they for? Aren't they separate ideas so how do the arrows link these ideas? Your comment has been taken into account and a major restructuring has been carried out in a number of areas. Another reviewer commented that the manuscript would be better understood if we included the questionnaire and for this reason it has been inserted, and we believe that the manuscript is better understood by inserting these questions. On the other hand, table 24 has been modified according to their comments and the arrows have been removed as they did not provide any information and might confuse the reader.

I feel that major rewrites will be necessary before it is up to an acceptable level for publication. Your comments have been taken into account and a major restructuring has been carried out in several sections, adding practical information and modifying certain aspects. We hope that this time the manuscript will be to your liking.

Reviewer 3 Report

Dear Authors,

even if some improvements were made, several issues related to the paper still need to be addressed.

- The use of English language is still very poor even if you said that it was extensively revised.

- I suggested to avoid redundant information in the Introduction section and those lines are still there (lines 35-36 and again at lines 46-47).

- There are several spelling mistakes ("ET" in the abstract), acronyms not explained at their first appearance in the main part of the manuscript ("TE" not explained, if you explain it in the Abstract you then have to do it again in the main text).

- Are all those table Appendix 1? I asked since it is mentioned in the text.

- All those tables put in Materials and Methods are way too confusing and should be reported as an appendix that should not be included in the main body of the manuscript.

- There are no number lines from the Results section and further on, making it difficult to detect any change 

 I suggested to divide the result section into sub-paragraphs, one for each research question, but this was not made. Results are difficult to interpret, and a Table should be made.

- You stated that a Table 3 was addedd but I did not find any Table 3.

I am not satisfied with the revised work.

An extensive English language revision is still needed. The Authors stated that the manuscript was revised by an "expert", but it was surely not enough.

Author Response

Dear Reviewer 3:

Thank you very much for your review and your time. We have carried out a thorough restructuring on the manuscript and your comments have been taken into account, you can check it in the attached document. 

Best regards and thank you.

even if some improvements were made, several issues related to the paper still need to be addressed.

- The use of English language is still very poor even if you said that it was extensively revised. Your comment has been taken into account and a review of the English content has been carried out by a professional, hopefully this time it will be optimal English for publication. Thank you.

- I suggested to avoid redundant information in the Introduction section and those lines are still there (lines 35-36 and again at lines 46-47). Your comment has been taken into account and the repeated sentences have been removed (information on lines (45-46) removed).

- There are several spelling mistakes ("ET" in the abstract), acronyms not explained at their first appearance in the main part of the manuscript ("TE" not explained, if you explain it in the Abstract you then have to do it again in the main text). Your comment has been taken into account and the wrong acronyms have been removed, it was a problem in the translation, sorry for the inconvenience and thank you.

- Are all those table Appendix 1? I asked since it is mentioned in the text. The appendix has been removed and inserted into the body of the manuscript, as commented by another reviewer in the previous round of revision. Thank you.

- All those tables put in Materials and Methods are way too confusing and should be reported as an appendix that should not be included in the main body of the manuscript. The appendix has been removed and inserted into the body of the manuscript, as commented by another reviewer in the previous round of revision. Thank you.

- There are no number lines from the Results section and further on, making it difficult to detect any change Your comment has been taken into account and this problem has been solved, apologies and thank you very much.

-  I suggested to divide the result section into sub-paragraphs, one for each research question, but this was not made. Results are difficult to interpret, and a Table should be made. Thank you very much for your comment in the results section, but we understand that the manuscript has many sections and we believe it is not necessary to divide the results further. In order to have an overview and as a summary, table 24 has been modified to present the results obtained from the consensus by the different experts, thank you.

- You stated that a Table 3 was addedd but I did not find any Table 3. The number of tables have been modified, please check the new version in its entirety, thank you.

I am not satisfied with the revised work. The introduction, methods and results have been thoroughly restructured. We hope you will be pleased with this new version. We apologise for the inconvenience.